# A photoelectrochemical platform for the capture and release of rare single cells

Stephen G. Parker[1,2,3], Ying Yang[1], Simone Ciampi[1], Bakul Gupta[1,2,3], Kathleen Kimpton[2,3,4], Friederike M. Mansfeld [2,3,4,5], Maria Kavallaris [2,3,4], Katharina Gaus[2,6,7] & J. Justin Gooding[1,2,3]

For many normal and aberrant cell behaviours, it is important to understand the origin of cellular heterogeneity. Although powerful methods for studying cell heterogeneity have emerged, they are more suitable for common rather than rare cells. Exploring the heterogeneity of rare single cells is challenging because these rare cells must be first preconcentrated and undergo analysis prior to classification and expansion. Here, a versatile capture & release platform consisting of an antibody-modified and electrochemically cleavable semiconducting silicon surface for release of individual cells of interest is presented. The captured cells can be interrogated microscopically and tested for drug responsiveness prior to release and recovery. The capture & release strategy was applied to identify rare tumour cells from whole blood, monitor the uptake of, and response to, doxorubicin and subsequently select cells for single-cell gene expression based on their response to the doxorubicin.

[1] School of Chemistry, The University of New South Wales, Sydney, NSW 2052, Australia. [2] Australian Centre for NanoMedicine, The University of New South Wales, Sydney, NSW 2052, Australia. [3] ARC Centre of Excellence in Convergent Bio-Nano Science and Technology, Melbourne, VIC 3052, Australia. [4] Children's Cancer Institute, The University of New South Wales, Sydney, NSW 2052, Australia. [5] Monash Institute of Pharmaceutical Sciences, Monash University, Melbourne, VIC 3052, Australia. [6] EMBL Australia Node in Single Molecule Science, School of Medical Sciences, The University of New South Wales, Sydney, NSW 2052, Australia. [7] ARC Centre of Excellence in Advanced Molecular Imaging, The University of New South Wales, Sydney, NSW 2052, Australia. These authors contributed equally: Stephen G. Parker, Ying Yang. Correspondence and requests for materials should be addressed to J.J.G. (email: justin.gooding@unsw.edu.au)

The importance of single-cell assays is that they reveal the diversity of cellular behaviour. Single-cell data is far richer than the common averaging of data from measurements from ensembles of cells. Knowledge of cellular heterogeneity can, for example, reveal whether the overall outcome of a treatment is caused by a common cellular response or by a range of responses[1]. Indeed, the overall outcome may be caused by aberrant rare cells where such behaviours might be masked in ensemble measurements[2]. To identify, and help understand aberrant behaviour, it would be ideal if single-cell technologies not only have the ability to identify phenotypically rare cells but also reveal the functional diversity of these cells. Examples of functional diversity from heterogeneity in rare cells include adult stem cells, which are believed to be responsible for observed variations in the efficiency of tissue repair[3, 4], maternal vs foetal cells, that have been postulated to play a role in the variations in immune response that mothers exhibit before and after child birth[5, 6] and circulating tumour cells (CTCs), where some, but not all, CTCs form metastatic tumours[7, 8]. The unmet need is assay methods that can capture rare cells, enable the investigation of single cells and allow the subsequent selection of individual cells for expansion and further study. Such methods would greatly enhance our understanding of the importance of heterogeneity in such rare cells.

Technologies have been developed for the isolation and manipulation of single cells from within a cell population. Examples include flow cytometry, micromanipulation or encapsulating single cells within a microwell, water droplet or a dielectrophoretic cage[2, 3, 9, 10]. As powerful as these techniques are, they are not well suited for analysing the heterogeneity amongst exceedingly rare cells. This is because either the likelihood of capturing enough rare cells is low or, with high throughput techniques, determining whether a rare event is the rare cell or noise can be problematic[11]. For example, if these single-cell isolation techniques were used to further understand the functional effects of the rare adult stem cells, rare foetal and maternal cells or rare CTCs within a complex sample, the unsynchronised nature of the much more abundant contaminating cells could hide any functionally relevant information obtained from the rare cells within the sample. A way to overcome this is to pre-concentrate these rare cells from contaminating cells. Technologies that can pre-concentrate and enumerate a subtype of rare cells from a sample containing mixed cells typically exploits morphological differences in these rare cells; most commonly size or the upregulation of specific surface antigens within the rare cells[12]. Such approaches regard all of the rare cells captured as identical as they use one set of markers to isolate these cells. To then explore the heterogeneity of these rare cells requires them to be addressed individually. Depending on the assays to be performed on these cells, exploring cell heterogeneity may require individual cells to be isolated, released and cloned.

Releasing a large number of cells captured on a surface has been achieved by applying an external stimulus, such as light, changing temperature, electrical potential or enzymatic release[10, 13–17]. If

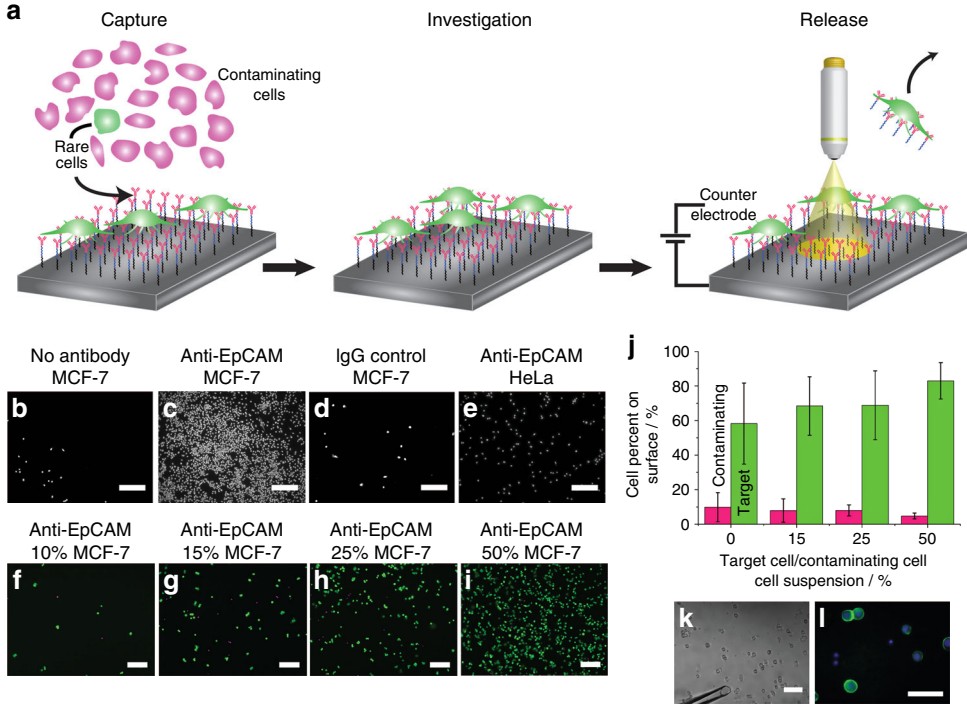

**Fig. 1** Capture performance of the antibody-modified silicon surface. **a** The capture and release device. Rare cells (green) are pre-concentrated on an electrochemically cleavable antibody-modified poorly doped p-type silicon surface from a mixture of cells (left). The pre-concentrated cells can be simultaneously analysed, for example with fluorescence microscopy (middle) before a unique single cell is selected and electrochemically released from the surface (right). **b**–**e** Fluorescence micrographs of live MCF-7 cells plated on an oligo(ethylene oxide)-terminated surface (**b**), an anti-EpCAM-modified surface (**c**), on a surface with non-specific donkey anti-mouse IgG antibody (**d**) and HeLa cells on an anti-EpCAM-modified surface (**e**). Cells were stained with the nuclear dye Hoechst 33342. Scale bar = 200 μm. **f**–**i** Fluorescence microscopy images of MCF-7 cells spiked into HeLa cells at 10% (**f**), 15% (**g**), 25% (**h**) and 50% (**i**) MCF-7 cells of total cells. MCF-7 cells were stained with calcein (green) while HeLa cells were stained with a Hoechst 33342 nuclear stain (magenta). Scale bar = 100 μm. **j** Quantitation of MCF-7 and HeLa cell densities on the silicon surfaces as a function of their composition in suspension. Error bars represent the margin of error at 95% confidence, n = 6. **k**–**l** MCF-7 cells were spiked (1% of total cells) into whole human blood before being captured on an anti-EpCAM-modified electrochemically cleavable silicon surfaces. All mononuclear cells were stained with the Hoechst 33342 nuclear dye (blue) whilst EpCAM-positive MCF-7 cells were also stained with FITC-conjugated anti-EpCAM antibodies (green). Scale bar in **k** = 100 μm; scale bar in **l** = 50 μm

these surfaces were used with the rare cells, then the further exploration could only be possible on an ensemble number of rare cells. Performing the further analysis on the stem cells, for example, would highlight the potential reasons for the observed variation in tissue repair but it would not reveal whether these differences are as a result of an equal contribution of all cells within the population or are dominated by a select few cells within the population. For this reason, it would be advantageous to be able to release only one cell. One way to release single cells could be to use unique surface chemistry that employs an electrochemically cleavable moiety[15, 18] and a novel electrochemical method we developed, called light-activated electrochemistry[19]. Light-activated electrochemistry uses semiconducting electrodes in depletion where any region on a monolithic surface can be made electrochemically active by shining light on that region. This removes the constraint of wires connecting specific locations on a surface.

To control the release of a single cell, however, is a considerable challenge that requires two stimuli so that the cell can first be observed and then released when desired. Herein we use a combination of light and electrochemical potential. Using light alone to release a single cell would mean visualising the cell would also release it. Using potential alone would not confine the release to a single cell. As such we visualise the surface and then reduce the beam size of the light to just one cell where upon potential is applied and thus a single cell could be released (Fig. 1a). We present the development of a capture & release surface that can firstly capture rare cells on the surface using antibodies for cell-surface markers, and also allow the surface to be imaged with a microscope, and eventually release any single cell on that surface when desired. The release of a single cell is achieved by using light-activated electrochemistry and a completely unpatterned surface that is prepared for the capture of the rare cells. Individual cells can be released when both light and electrochemical potential are applied. Selective cell release can be thus achieved since the potential is confined to a given location by light (Fig. 1a). This means that preliminary information about the heterogeneity amongst rare cells, such as drug response, can be obtained before selecting the single cell of interest to be released and further explored.

## Results

**Capturing a subtype of rare cells**. The entire technology is facilitated by the ability to modify p-type silicon (100) surfaces with a custom designed capture & release surface. The surface is composed of four layers (Supplementary Fig. 1). First, a passivating organic monolayer derived from 1,8-nonadiyne was attached to a hydrogen terminated p-type silicon surface via thermal hydrosilylation[20, 21]. This base monolayer served two purposes. The first was in preventing oxidation of the silicon in an aqueous solution even under applied potentials[21]. This is vital as any oxidation of the silicon would preclude it functioning as an effective electrode. Secondly, the distal alkyne of the monolayer provides a convenient moiety for further coupling reactions. The second layer was formed by the attachment of an electrochemically cleavable linker[15, 18, 22] molecule to the passivating monolayer via a copper-catalysed azide-alkyne cycloaddition reaction. In order to prevent fouling and enable pre-concentrating of rare cells, the third layer of the surface was an oligo(ethylene oxide) species that resists non-specific adsorption of proteins and cells from biological fluids. The final layer was antibodies that selectively bind to surface antigens on rare cells to be captured and can be bioconjugated to the antifouling layer. Each step in the modification procedure was characterised with X-ray photoelectron spectroscopy and X-ray reflectometry as shown and described in the supplementary information (Supplementary Fig. 2).

The defining element of the surface chemistry is the custom designed and synthesised cleavable linker (Supplementary Fig. 3 & 4), a quinone species that can be reduced to a cyclised hydroquinone via a so-called trimethyl lock lactonisation system[15, 18, 22]. This cleavage reaction then results in the release of the attached species at the distal end of the cleavable moiety (Supplementary Fig. 1). After binding the cleavable molecule to the surface, it was chemically oxidised to reveal a terminal carboxylic acid moiety. The carboxylic acid was activated using carbodiimide coupling to give a succinimidyl ester that allows coupling of a protein-resistant antifouling layer on top of the cleavable layer. In this case, oligo(ethylyne) oxide molecules[23] containing six repeating units (OEO6) and an amine at the proximal end to attach to the succinimidyl ester on the surface was used. Finally, antibodies, to pre-concentrate the target cells, were bioconjugated to the OEO6 species via a N,N-disuccinimidyl carbonate activating group. To monitor the conjugation of the antibodies to the OEO6 layer, a Cy3-tagged antibody was tethered to the surface. MCF-7 cells were chosen as the model for the cell capture & release system. To selectively capture cells, anti-epithelial cell adhesion molecule (EpCAM) antibodies that recognises the transmembrane protein, EpCAM, which is over-expressed on MCF-7 cells and many other cancer cells (Supplementary Fig. 5) was used.

To test whether the capture & release surface is selective for MCF-7 cells, surfaces with and without anti-EpCAM antibodies were incubated with MCF-7 cells (Fig. 1b, c). The surface without anti-EpCAM in Fig. 1b showed a cell density of $25 \pm 8$ cells mm$^{-2}$, (95% confidence interval; $n = 6$) while the surface with the anti-EpCAM had $1310 \pm 68$ MCF-7 cells mm$^{-2}$ (Fig. 1c). Further, a surface with an incorrect antibody (a non-specific IgG antibody) only captured $12 \pm 4$ MCF-7 cells mm$^{-2}$ and when an EpCAM-deficient cell line (HeLa cells) was used[24], the cell density was $42 \pm 15$ cells mm$^{-2}$ (Fig. 1d, e). These results demonstrated the efficacy of the oligo(ethylene oxide) moieties in the monolayer to resist non-specific cell adsorption and provide selectivity for cells with upregulated EpCAM expression. To further show the capability of the capture & release surface to selectively capture a given subtype of cells from a mixed sample, surfaces were exposed to a mixture of MCF-7 and HeLa cells (10–50% MCF-7 cells of the total number of cells) (Fig. 1f–i). As shown in Fig. 1j, the surfaces were 6–8-fold more selective for MCF-7 cells than HeLa cells. A closer inspection of the morphologies of each cell type on the surface revealed that MCF-7 cells spread and formed filopodia whilst HeLa cells maintained a spherical shape on the surface (Supplementary Fig. 6). This shows that the MCF-7 cells had selectively bound to the antibodies on the surface whilst the HeLa cells did not. Next, we tested that the device could be used in complex biological samples. As such, whole human blood was spiked with 1% MCF-7 cells (calculated from a blood cell concentration of $5 \times 10^9$ cells mL$^{-1}$ [25]) and exposed to anti-EpCAM-modified capture surfaces (Fig. 1k). The surface had high selectivity for the MCF-7 cells with only a few blood cells (the smaller cells in Fig. 1k) adhering to the surface. Immunostaining with a FITC-conjugated anti-EpCAM antibody confirmed that the larger cells were MCF-7 cells (Fig. 1l). Finally, the capture capabilities of the device on lung tumour cells from a complex sample was evaluated. To test this, EpCAM-positive non-small cell lung cancer (NSCLC) cells, isolated from a murine cancer model, were spiked into whole human blood. Supplementary Movie 1 shows that the larger lung tumour cells were strongly tethered to the EpCAM-modified silicon surface whilst the smaller contaminating blood cells were loosely adhered and easily removed. The ability to selectively capture a subtype of cells out of whole blood illustrates the capability of the device to pre-concentrate rare diseased cells. This ability ensures that our

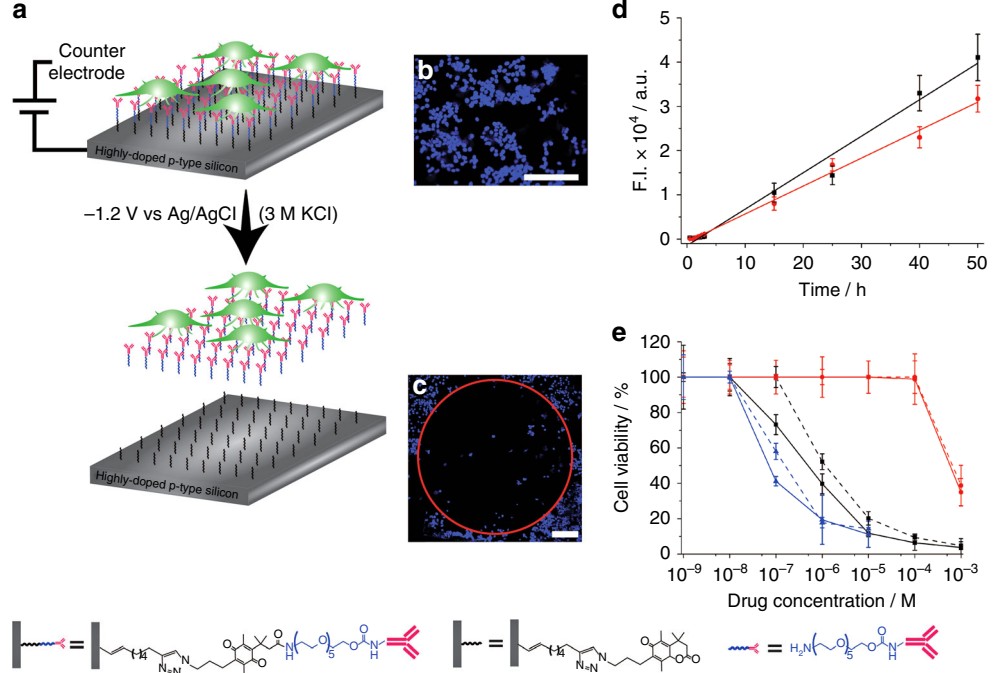

**Fig. 2** Ensemble electrochemical release of cells and their viability. **a** Cell capture & electrochemical release of all adherent cells. **b** MCF-7 cells captured on an anti-EpCAM-modified p-type silicon surface (0.001–0.003 Ω cm), modified as shown (bottom left), and stained with Hoechst 33342 (blue). **c** The MCF-7 cells from **b** were removed from the central region where an o-ring confined the electrolyte (DMEM with 10% foetal bovine serum) to this region. MCF-7 cells outside of this region were not exposed to the electrolyte and were therefore not released from the surface. Scale bars = 200 μm. **d** Respiration, measured with alamarBlue®, of captured & released MCF-7 cells over 50 h (red circles and lines), compared to an identical number of cultured MCF-7 cells (black squares and lines). **e** Cell viability and response of captured & released MCF-7 cells (dotted lines) and cultured MCF-7 cells (solid lines) to doxorubicin hydrochloride (blue lines), 5-fluorouracil (black lines) and capecitabine (red lines). Error bars indicate margin of errors at 95% confidence, $n = 6$

understanding about the effects of cellular heterogeneity amongst these rare cells are not as a result of noise. In order to exploit the full range of techniques available to explore cellular heterogeneity amongst these rare cells, they are required to be assessed, and individually selected and released from the surface in a manner that does not affect their viability or phenotype.

**Photoelectrochemical releasing viable selected single cells.** Next, the ability to release of the cells from the surface was demonstrated and whether the release mechanism affected the cell function was investigated. For this purpose, the cell release was performed on the ensemble level where the entire layer of captured cells was released from the modified silicon electrode. In this case the silicon electrode was highly doped such that it behaved in a similar manner to a metal electrode. Applying −1.2 V vs Ag|AgCl for 240 s to the electrochemically cleavable surface led to the release of 82 ± 5% (95% confidence interval, $n = 6$) of the attached MCF-7 cells (Fig. 2a–c) (optimisation polarisation is described in Supplementary Fig. 7 & 8). The linker length between the cleavable layer and the silicon surface will not affect the applied potential for cell release as has been demonstrated by Gooding et al.[26]. Preparing an equivalent surface, but with a non-electrochemically cleavable moiety (Supplementary Fig. 4) led to the removal of only 1 ± 0.3 % of the captured cells under the same conditions (Supplementary Fig. 9). This indicates the removal of the captured cells was due to the lactonisation of the electrochemically switchable molecule cleaving the monolayer below the cells.

The effect of electrochemical release on cells was examined using trypan blue exclusion test.[27, 28] The MCF-7 cells post capture & release maintained a viability of 90 ± 2.7% (95% confidence interval, $n = 6$) while the MCF-7 cells in culture

exhibited viability of 92 ± 7.2%, strongly suggesting that viability was not affected by the capture & release process. In particular, the integrity of the cell membrane, which can rupture in electrochemical processes such as electroporation[29], did not appear to be affected. The captured & released cells were able to form continuous cultures after a 15-day period. Cultured MCF-7 cells, at equivalent concentrations to the captured & released MCF-7 cells, displayed similar replication rates as the captured & released MCF-7 cells (Supplementary Fig. 10), highlighting that the capture & release process did not affect growth of the cells despite exhibiting slightly slower initial respiration rates (Fig. 2d).

Subsequently, the response of the captured & released cells to commonly used chemotherapeutics, compared to cultured cells, was explored (Fig. 2e). Measuring the cytotoxicity revealed that the captured & released and the cultured MCF-7 cells had $IC_{50}$ values of $1.9 \times 10^{-1} \pm 6.0 \times 10^{-2}$ and $8.0 \times 10^{-2} \pm 5.8 \times 10^{-3}$ μM (95% confidence interval, $n = 4$) to doxorubicin hydrochloride, respectively. It was also observed that 5-fluorouracil had a similar effect on the MCF-7 cells with $IC_{50}$ values of 1.0 ± 0.35 and 0.50 ± 0.19 μM for the captured & released and cultured MCF-7 cells, respectively. The prodrug, capecitabine had $IC_{50}$ values of $7.0 \times 10^1 \pm 5.7 \times 10^1$ and $6.5 \times 10^1 \pm 1.5 \times 10^1$ μM for the captured & released and cultured MCF-7 cells, respectively. The high $IC_{50}$ values for capecitabine highlight that, as expected, this non-toxic prodrug did not have a major effect on both the cultured and the captured & released MCF-7 cells. The similar $IC_{50}$ values for the MCF-7 cells after capture & release (when compared to the $IC_{50}$ values before capture & release for different drugs) highlight the minimal effect the process has on the response the cells have towards chemotherapeutics and alludes to the capabilities that this device has towards personalised treatment.

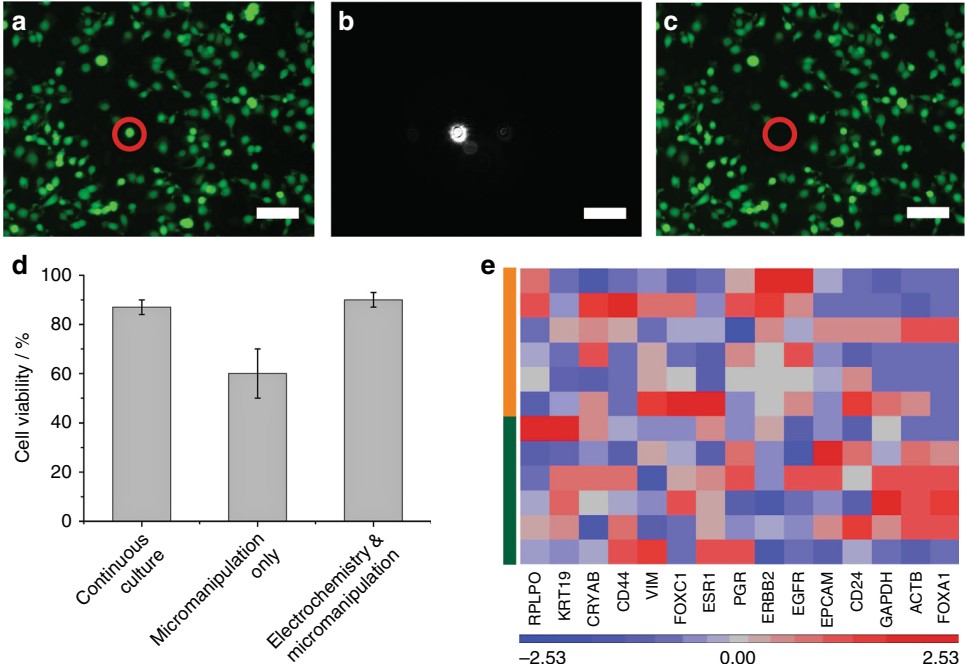

**Fig. 3** Localised release of single cells and investigating the release effect on cells. **a–c** Fluorescence microscopy images of calcein-stained cells (green) before release (**a**); during electrochemical "cleavable" reaction to a specific area by illuminating a single cell with a laser (**b**); after release and recovery with a micromanipulator (**c**). Scale bar = 100 μm. **d** Cell viabilities of MCF-7 cells in continuous culture that have been left in ambient conditions for 1 h (left column), recovery from an anti-EpCAM-modified silicon sample using the micromanipulator without electrochemical release (middle column) and recovered from a switchable anti-EpCAM-modified silicon surface with a microcapillary after electrochemical release of the cells (right column). Error bars represent the margin of errors at 95% confidence, $n = 6$. **e** The genetic expression of commonly used epithelial cancer biomarkers for six single electrochemically isolated MCF-7 cells (orange) and six single MCF-7 cells obtained by serially diluting the continuous culture followed by recovery with the micromanipulator (green). The degree of upregulation or downregulation relative to the serially diluted MCF-7 cells is represented with a red or blue box, respectively

In summary, it is demonstrated that the capture & release surface was able to release cells without compromising the integrity and response of the cells. The next steps in the capture & release strategy was to use poorly doped silicon surface such that light-activated electrochemistry could be used for the selective release process of a single cell.

Light-activated electrochemistry was employed to localise the electrochemical release to single cells. To achieve this, required a switch to a poorly doped p-type silicon surface (10–20 Ω cm) modified with the capture & release surface chemistry. The switch to poorly doped silicon was a prerequisite for being able to release a single cell as, at the potential required to cleave the electrochemically cleavable unit the silicon is in depletion. As such, in the dark the electrode is not sufficiently conducting for electrochemical reactions to take place[19]. Illuminating the poorly doped silicon promoted electrons from the valence band to the conduction band, which increases the conductivity and hence allows the electrochemical cleavage reaction to proceed. An optical microscope, which was used to observe the cell morphologies on the surface, could now activate the silicon surface for electrochemistry. In this way, it was possible to compare the cell morphology of the captured cells prior to their localised electrochemical release, either under brightfield microscopy or in conjunction with fluorescence labelling (Fig. 3a).

In order to limit the beam size to one cell, the microscope was configured with an optical fibre with a tunable beam size from 10–100 μm such that a single cell could be illuminated (Fig. 3b). A potential of −1.2 V vs Ag|AgCl (3 M KCl) was applied to the surface for 240 s while the single cell was illuminated. The cleavage of the electrochemical linker under the illuminated regions meant that the cell of interest was easily collected by a microcapillary pipette. Critically, other cells in the dark area were still bound to the surface well (Fig. 3c). Supplementary Movie 2 illustrates how the strong adhesion of the MCF-7 cells to the silicon surface prevented their recovery with the micromanipulator prior to localised electrochemical release and how specific the light-induced electrochemical process is to only the illuminated cell while non-illuminated cells remained attached. Supplementary Movie 3 shows that a spatial resolution of 30 μm can be obtained with this device.

As with the ensemble release, it was necessary to show that the release of a single cell of interest had no significant impact on the cell. Cell viabilities were evaluated by a trypan blue exclusion test[27, 28] for the MCF-7 cells after their light-activated electrochemical release and recovery with a micromanipulator (Fig. 3d). MCF-7 cells that underwent light-activated electrochemical release remained 90 ± 3% (95% confidence interval, $n = 4$) viable (compared to 87 ± 3% for MCF-7 cells left for 1 h the tissue culture flask). In contrast, mechanically removing cells from the surface with the microcapillary pipette lead to a cell viability of 60 ± 10%. The impact of the localised release on gene expression levels was also assessed. A total of 20,818 genes from six electrochemically released single MCF-7 cells were compared with six single cultured MCF-7 cells. It was discovered that only 18 of the 20,818 genes investigated displayed a difference in expression from cells between treatments (minimum of twofold increase with a $p$-value ≥ 0.01, one-way ANOVA). Gene ontology revealed that these genes were not involved with processes known to be involved with the cancer pathway. Furthermore, particular attention was paid to the expression of commonly used epithelial cancer biomarkers amongst the MCF-7 cells (Fig. 3e). It was discovered that variations within a pool of six cells, whether they

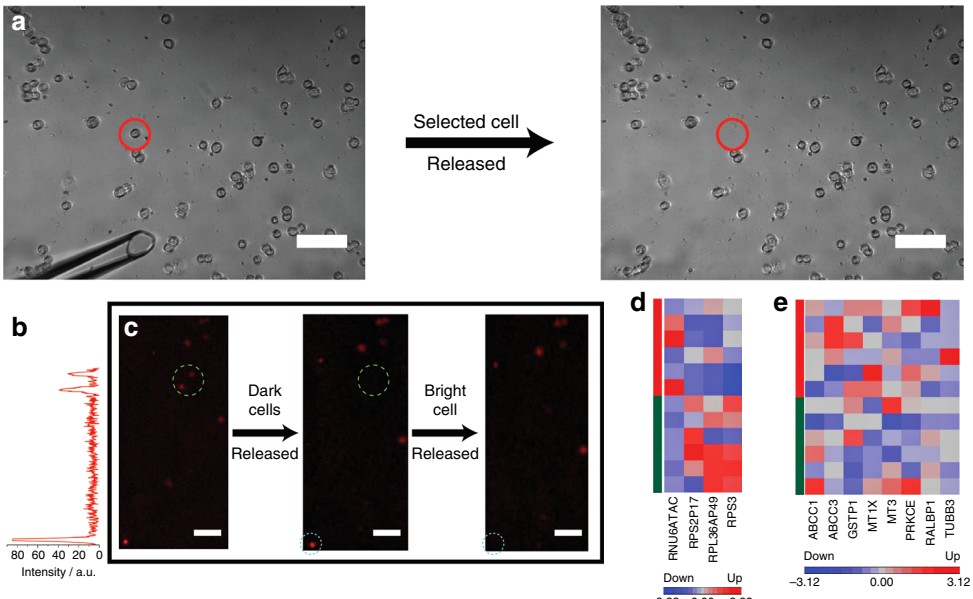

**Fig. 4** Localised release of single MCF-7 or lung tumour cells from blood sample. **a** Brightfield images of MCF-7 cells (1%) spiked into whole human blood and captured on an anti-EpCAM-modified electrochemically cleavable silicon surface (left) prior to electrochemical release and recovery of a selected single MCF-7 cell and (right) after electrochemical release (scale bar = 100 μm). **b** Intensity plot of lung tumour cells captured on an anti-EpCAM-modified electrochemically cleavable silicon surface and treated with doxorubicin hydrochloride after being spiked into whole human blood. **c** Fluorescence micrographs of lung tumour cells treated with doxorubicin hydrochloride (10 μM), showing the ability of the device to isolate a single cell based on its doxorubicin hydrochloride uptake. Scale bar = 50 μm. **d** The expression of genes displaying a twofold change with a $p$-value ≥ 0.05, one-way ANOVA, between lung tumour cells that have a high (red) and a low (green) DOX uptake and **e** reported to play a role in DOX-resistance between lung tumour cells that have a high (red) and a low (green) DOX uptake. The degree of upregulation or downregulation is represented with a red or blue box, respectively

be captured & released or just cultured, was greater than the variation between the two pools. This provides evidence that the localised individual cell release process, like the ensemble release, did not affect the genotype of the cells. As such it is expected that information obtained by analysing single cells using this technique can be attributed to cellular heterogeneity rather than to effects caused by the capture & release process.

After determining that the localised release process does not have an effect on the genetic expression of individually recovered cells, the performance of the device with rare cells and complex samples was determined. The localised release process was first successfully used to recover an individual MCF-7 cell that had been spiked into whole human blood (Fig. 4a). The capabilities of the device were extended by transitioning from spiking cell lines to lung tumour cells into blood and showing that the rare lung tumour cells could be interrogated prior to releasing and recovering one. The latter was illustrated by treating the captured lung tumour cells with doxorubicin hydrochloride (10 μM). Figure 4b reveals variations in the uptake of doxorubicin hydrochloride amongst the lung tumour cells. Figure 4c illustrates the capability of the technique to sequentially release individual lung tumour cells that have a high doxorubicin-uptake or low doxorubicin-uptake (as indicated by a minimum of a twofold difference in the mean intensity of cell-associated doxorubicin). The single-cell gene expression of six single-lung tumour cells with high doxorubicin-uptake (with a minimum mean intensity of 80 A.U.) and six single-lung tumour cells with low doxorubicin-uptake (with a maximum mean intensity of 40 A.U.) revealed upregulation (minimum of twofold increase with a $p$-value ≥ 0.05, one-way ANOVA) of genes encoding for proteins within the ribosome[30] (Fig. 4d) within lung tumour cells with low doxorubicin-uptake and alludes to the enhanced initiation of translation as a factor towards doxorubicin resistance. Interestingly, as shown in Fig. 4e, no discernible difference was observed for

genes encoding for proteins known to affect doxorubicin resistance[31–35] between lung tumour cells with high doxorubicin-uptake and low doxorubicin-uptake. The single-cell gene expression highlights the capability of the device to discern biological information that would otherwise be masked using ensemble techniques.

## Discussion
In this study, a concept for the capture, characterisation and release of single cells using a smart surface is reported. The surface is a silicon surface modified with a self-assembled monolayer (SAM) consisting of four different layers, fabricated in a stepwise manner to impart four different functionalities; a layer to prevent oxidation of the silicon surface, an electrochemically cleavable unit, chemistry to resist non-specific adsorption of cells and antibodies to selectively bind cells of interest. In this report, the cells of interest are a model for CTCs, MCF-7 breast cancer cells and lung tumour cells from a subcutaneous non-small cell lung cancer model that both express EpCAM. It is demonstrated that cells can be selectively captured on the anti-EpCAM-modified silicon surface, even from whole blood, and when desired, photo-electrochemically released from the modified silicon surface. This advance is based on the photo-resolved electrochemical property of a silicon substrate[19] and the attached electrochemically cleavable trimethyl lock[36]. The capture & release surface is compatible with imaging and allows direct visualisation of cell morphology and enables localising the release of any single cell the user desires. Furthermore, the capture & release process was shown to have no impact on cell viability, differentiation or proliferation. Importantly, the modular nature of the surface chemistry means a simple change of the antibody, or using a combination of antibodies, will enable the smart surface concept to be able to capture & release other cell types.

The requirement for two stimuli, visible light and potential, to release the cells is a key aspect of this technology. Visible light allows cells to be viewed to determine which cells to release. However, if light alone was the stimulus, then viewing the cells under a microscope would cause them to be released. Similarly, the use of a shorter wavelength of light to stimulate the release after viewing, would risk causing photo-damage to the cells. Using potential alone would result either in all cells being released or the requirement for many wires and bonding pads leading to many independently addressable electrodes. Having wires and bonding pads for each cell would consume considerable space on the chip, and thus limit the number of cells that could be captured and released. Furthermore, the surface would need to be engineered so the cells are captured only where the electrodes are located. The electrochemical method, referred to as light-activated electrochemistry[19], has none of these constraints. Light-activated electrochemistry simply requires a microscope and a standard electrochemical potentiostat. The capture & release surface is entirely unpatterned such that cells can be released from anywhere they are attached to the surface with no limits on their density other that the space being available to capture the cells of interest. The cells can be viewed under the microscope and only released upon confining the light source to the area of interest, by reducing the microscope aperture, and applying the potential. The cells are only released at the site of illumination as only these sites have sufficient charge carriers in the silicon for the substratum to be conducting. Hence the behaviour of each cell on the surface can be determined prior to selecting a unique cell to release, which could then be further characterised or cloned to better understand its relative functional contribution to the progression of the disease. Microfluidics could also be coupled with this device to improve the speed to isolate a single cell. An example of the benefit of the capabilities exhibited by this device is the ability to determine the response of each rare cell has towards a drug before determining the contribution that this cell has towards the progression of the disease. This has already been demonstrated herein with the commonly used doxorubicin hydrochloride (Fig. 3). This technology could be used for drugs that are still in clinical trials to more accurately determine the efficacy of the drug as it will provide insight into the number of targeted cells that are affected by the drug and the extent of which each cell is affected.

Furthermore, the capture & release surface was designed so it can be used in combination with microfluidic or magnetic pre-concentration systems such that the surface can be used for capturing rare cells within a sample. This means that single CTCs can be isolated, grown into three dimensional spheroids for downstream genetic and phenotypic analysis including injecting into mice to determine their tumourigenic and metastatic potential.

Finally, the ability to first identify functional or morphological features of individual cells prior to deciding which cell to release provides the surface with a level of flexibility that is quite unique. For example, although the surface needs to reduce non-specific cell adsorption, it does not need to completely prevent it as the identification of any nonspecifically adsorbed cells means that such a cell can just be left on the surface while other cells of interest are released. It also means that the surface can be modified with antibodies for different cell-surface markers such that heterogeneity in the expression of cell-surface markers does not cause a subpopulation of cells to not be captured.

Compared with the commonly used single-cell sorting methods, the approach presented herein allows cells to be selected based on morphological, surface expression or response to drugs and stimuli. This system could be a powerful tool to resolve cell heterogeneity after transforming the phenotype differences into genetic information, and further determine whether the differences are random or meaningful, and optimise the models of individual cell behaviours.

## Methods

**Chemicals**. All chemicals, unless noted otherwise, were of analytical grade and used as received. MilliQ™ water (18 MΩ cm) was collected through a Millipore™ water purification system and was used to prepare solutions and for chemical reactions. Acetone, dichloromethane, diethyl ether, ethanol, ethyl acetate, N,N-dimethylformamide, light petroleum (60–80 °C), 2-propanol and toluene for surface cleaning, chemical reactions and purification procedures were redistilled prior to use. Dry acetonitrile was obtained from PureSolv MD 7 Solvent Purification System (Innovative Technology Inc., Galway, Ireland). Hydrogen peroxide (30 wt % in water, Sigma-Aldrich), hydrofluoric acid (J.T. Baker) and sulphuric acid (Sigma-Aldrich) were of semiconductor grade. 1,8-nonadiyne **1** (Sigma-Aldrich) was redistilled from sodium borohydride under reduced pressure (60 °C, 29 Torr) and stored under a dry argon atmosphere prior to use. Chemicals required for the synthesis of the electrochemically switchable molecule are described in the supporting information. Copper(II) sulphate hexahydrate was purchased from Ajax Finechem. Sodium L-ascorbate, N-hydroxysuccinimide and N,N'-disuccinimidyl carbonate (DSC) were purchased from Sigma-Aldrich, N-bromosuccinimide and 1-(3-dimethylaminopropyl)-3-ethylcarbodiimide hydrochloride (EDC) were purchased from Alfa Aesar. OEO6 was purchased from Biomatrik. Dulbecco's phosphate buffer saline (DPBS) was used for biological applications and was filtered before use. 1× trypsin was diluted from 10× trypsin stock using 1× DPBS. Growth medium was prepared by mixing Dulbecco's modified eagle medium (DMEM) (440 mL), foetal calf serum (FCS) (50 mL), L-glutamine (5 mL) and penicillin/streptomycin mix (5 mL). Trypan blue, the LIVE/DEAD stain (calcein AM for the live component and ethidium homodimer-1 for the dead component), alamarBlue® and Hoechst 33342 were purchased from Invitrogen. Doxorubicin hydrochloride, 5-fluorouracil and capecitabine were purchased from Sapphire Bioscience. The Ovation® One-Direct System was purchased from Integrated Science.

**Purification and analysis of synthesised compounds**. Thin-Layer Chromatography (TLC), to check the purity of synthesised compounds, was performed on aluminium-backed silica gel sheets (Merck, 60F254, normal phase). Preparative silica gel chromatography, to purify synthesised compounds, was performed using silica gel (Grace GmbH, 40–63 μm, normal phase). Nuclear magnetic resonance (NMR) spectra, to confirm the identity of the synthesised compounds, were obtained on a Bruker Avance 300 spectrometer, using the solvent signal as the internal reference. Fourier-transform infrared (FTIR) spectra were recorded on a Thermo Nicolet Avatar 370 FTIR spectrometer by accumulating a minimum of 32 scans and selecting a resolution of 2 cm$^{-1}$. The electrochemically switchable linker molecule was synthesised as described in the Supplementary Fig.3.

**Silicon wafers**. Prime grade single-sided polished silicon wafers 100-oriented (<100> ± 0.5°), p-type (boron), 525 ± 25 μm thick, 0.001–0.003 Ω cm or 10–20 Ω cm resistivity were purchased from SILTRONIX.

**Antibodies and cells**. The primary antibodies, used for cell capture, anti-mouse anti-EpCAM [VU-1D9] (catalogue number: ab187372) were purchased from Abcam (Cambridge, MA, USA). The received stock solution was diluted 1:200 (v/v) in PBS immediately prior to use. FITC-conjugated anti-EpCAM antibodies [B29.1 (VU-1D9)] (FITC) (catalogue number: ab8666), used for immunostaining, were purchased from Abcam (Cambridge, MA, USA). The received stock solution was diluted 1:100 (v/v) in growth medium immediately prior to use. Cy-3-conjugated donkey anti-mouse antibodies (catalogue number: 715-165-150) (used to show the bioconjugation of antibodies on top of the OEO6 antifouling layer, as well as the specificity of the antibody-antigen interaction on the surface) were purchased from Jackson ImmunoResearch (West Grove, PA, USA). The stock Cy-3-conjugated donkey anti-mouse antibody solution was prepared by dissolving the antibody powder in glycerol (250 μL) and Milli-Q water (250 μL). The received stock solution was diluted 1:200 (v/v) in PBS immediately prior to use. MCF-7 and HeLa cell lines, used to test the ability of the surface to selectively isolate individual cells, was purchased from CellBank Australia (Westmead, NSW, Australia). H441 NSCLC cells, used for subcutaneous injections, was purchased from CellBank Australia (Westmead, NSW, Australia). All cell lines were tested with MycoAlert™ Mycoplasma Detection Kit (Lonza Group) and found to be mycoplasma negative. The MCF-7 cell line was authenticated at CellBank Australia using short tandem repeat (STR) profiling.

**Human blood**. Single donor human whole blood was purchased from innovative research™ (Novi, MI, USA) and treated with K2 EDTA to prevent coagulation.

**Statistical analysis**. Uncertainties depict the 95% confidence interval and were calculated using $\mu_{95\%} = s.t_{0.025}/\sqrt{n}$ where the degrees of freedom = $n - 1$. Comparison of two treatments, such as cultured vs captured & released was quantitated by setting up a null hypothesis, $H_0$, that the treatments were the same ($x_1 - x_2 = 0$) and an alternate hypothesis $H_a$, that the treatments were not the same ($x_1 - x_2 \neq 0$) where $x_1$ = the larger average $- \mu_{95\%}$ and $x_2$ = the smaller average $+ \mu_{95\%}$.

**Microscopy**. Fluorescence images were recorded on an Olympus BX53 upright microscope. Hoechst 33342 images were collected with a DAPI filter with an

excitation filter: 360 nm and an emission filter: 460 nm. FITC images were recorded with a FITC filter with an excitation filter: 470/40 nm and emission filter: 525/50 nm. Single-cell illumination was achieved using a white light laser (Thorlabs) with a 0.1 A laser current, a photodiode current of 0.0003 mA, thermopile voltage of 2.46 V and optical power of 0.02 mW. The spot size of the laser was adjustable to illuminate the spot of a single cell. The images were captured with an Olympus XM10 monochrome camera, recorded with the CellSens software (Olympus Australia, Notting Hill, Victoria, Australia) and processed with ImageJ 1.42q (National Institutes of Health, USA).

**Growth of subcutaneous lung tumours**. Female Balb/c nude mice (6–8 weeks of age) were obtained from the Animal Resource Centre, (Perth, Western Australia). All animal experiments were approved by the Animal Ethics Committee, University of New South Wales (ACEC no. 14/124B). Mice were inoculated subcutaneously into the flank with H441 NSCLC cells ($1 \times 10^6$). Tumours were measured twice weekly using digital calipers, and tumour volume was calculated using the formula $1/2$ ($L \times W^2$), where $L$ and $W$ are the length and width of the tumour, respectively. All mice were humanely killed once the tumours reached 1 $cm^3$.

**Dissociation of lung tumours**. Collected lung tumours were dissociated using a Miltenyi biotec GmbH gentleMACS™ Octo Dissociator with a human tumour dissociation kit (Miltenyi biotec GmbH). Red blood cells were removed from the tumour cells by suspending the pelleted tumour cell suspension in a 0.22 µm-filtered red blood cell lysis buffer (1 mM ammonium bicarbonate and 114 mM ammonium chloride in deionised water (10 mL)) for 5 min before centrifuging the suspension at $225 \times g$ for 5 min to pellet the purified tumour cells. The tumour cells were then resuspended in RPMI 1640 containing 10% FBS (2 mL) to be spiked into human blood.

**Preparation of MCF-7 or lung tumour cell-spiked suspensions**. For the preparation of MCF-7 cells in HeLa cells, the MCF-7 cells were first stained with calcein (1 µM) while the HeLa cells were stained with Hoechst 33342 (33 µM). The various cocktails were prepared by mixing various volumes of each cell suspension together (total number of cells was $6.57 \times 10^5$ cells in 2 mL media for all compositions). For the preparation of MCF-7 cells in whole human blood, $1.2 \times 10^6$ MCF-7 cells in growth medium (2 mL) were mixed with whole human blood (7.5 mL). For the preparation of dissociated lung tumour cells in whole human blood, $7.7 \times 10^5$ lung tumour cells in RPMI 1640 containing 10% FBS (2 mL) were mixed with whole human blood (7.5 mL).

**Capturing cells**. Cell suspensions were incubated on top of the anti-EpCAM-modified silicon surface for 15–25 min with gentle agitation. The surfaces were then gently washed with growth medium to remove loosely bound cells before leaving in growth medium for further analysis and electrochemical release.

**Electrochemically releasing cells**. A bio-logic SP-200 potentiostat (Lyon, France) was used to apply a negative bias vs Ag|AgCl (3 M KCl) to initiate electrochemical-switching of the switchable components within the SAM. A customised PTFE Teflon three electrode cell was used with a modified silicon surface as the working electrode, a platinum mesh as the counter electrode, and Ag|AgCl in 3 M KCl as the reference electrode. All potentials were indicated as $E$, and reported vs the reference electrode. Ohmic contact between the silicon substrate and a copper plate was ensured by rapidly rubbing a gallium indium eutectic onto a close series of marks (emery paper) aimed to expose the bulk of the silicon electrodes.

To determine the release efficiency, the growth medium above the surface was gently removed with an autopipette before enumeration of the remaining cells on the surface. The same process was done with a surface that did not contain an electrochemically cleavable unit to confirm that release was due to the cleavage process and not due to the handling of the cells.

For the determination of the viability of the released cells and the culturing of the released cells, the cells were recovered with a micromanipulator rather than a micropipette. This was to minimise the shear forces associated with the recovery of the cells and to maximise cell viability. The released cells (~15,000) were placed in a 0.2 mL Eppendorf tube for trypan blue counting and viability determination, diluted to 5000 cells mL$^{-1}$ for alamarBlue® viability determination or transferred to a T25 flasks to be placed at 37 °C in 5% CO$_2$ to promote cell proliferation.

**Counting and viability determination of released cells**. The released cells (10 µL) were mixed with an 0.4% trypan blue solution (Invitrogen, 10 µL) that had been filtered through a 0.22 µm filter. The mixture was then loaded into Invitrogen Countess™ Cell Counter Chamber slides (10 µL in the A and B port of the slide) and counted with the Invitrogen Countess™ automated cell counter.

**Respiration of released cells**. The respiration of released cells was measured via adding alamarBlue® (10% v/v of the volume medium that the released cells were suspended in) to the released cells and then aliquoting them into a 96-well plate (200 µL per well) to measure their fluorescence intensity at different time intervals. The fluorescence intensity was measured using a BMG LABTECH FLUOstar

omega with an excitation wavelength of 544 nm, an emission wavelength range of 590–610 nm and a gain factor of 1000 and was compared to a suspension of MCF-7 cells with the same concentration and volume that had not undergone the capture & release process. A background sample prepared by adding alamarBlue® (10% v/v) to growth medium not containing cells was deducted from the cultured and captured & released cell samples.

**Drug assay on ensemble released MCF-7 cells**. Recovered MCF-7 cells (~15 000 MCF-7 cells) were placed in a T25 tissue culture flask and diluted with recovery medium (DMEM containing 30% FCS) and left for 15 days, with the MCF-7 cells being trypsinised and reseeded after 11 days to remove formed clusters. Once the released MCF-7 cells had reached 80% confluency, they were split and resuspended in growth medium. Before undertaking the drug assays, a growth profile of MCF-7 cells was carried out in order to determine the plating densities and time lengths required in order for the cells to be in a logarithmic stage of growth throughout the entire duration of the experiment. MCF-7 cells (200 µL) were placed in 96-well plates at various concentrations (334, 683, 1047, 1471, 2668 cells per well). AlamarBlue® (40 µL) was added to each well containing MCF-7 cells and the fluorescence intensity was measured daily using a gain factor of 750. It was determined that 1000 MCF-7 cells per well maintained this logarithmic phase of growth for 7 days (Supplementary Fig. 11) and therefore this plating density was suitable for the drug assays. The captured & released and cultured (on the same passage number as the captured & released MCF-7 cells) MCF-7 cells were left for 1 day after plating them (1000 cells per well, 250 µL) in a 96-well plate. At this stage, the growth medium was aspirated and replaced with either capecitabine, doxorubicin hydrochloride or 5-fluorouracil ($10^{-3}$ M -$10^{-9}$ M in growth medium, 250 µL) and left for 5 days. AlamarBlue® (25 µL) was then added to each well and left for 9 h before measuring the fluorescence intensity with an excitation wavelength of 544 nm, an emission range of 590–610 nm and a gain of 750. To determine the % cell viability, the fluorescence intensity of each concentration of each drug was compared to the cultured MCF-7 cells or captured & released MCF-7 cells with no drug added.

**Single diseased cell isolation**. The anti-EpCAM-modified silicon surface was loaded into a customised electrochemical chamber designed to fit on the microscope stage. The customised electrochemical chamber contained a copper base such that an electrical contact between the potentiostat and the silicon surface was established. MCF-7 cells ($3 \times 10^4$ cells mL$^{-1}$, 4 mL) or the MCF-7-spiked whole human blood (7.5 mL) was loaded into the customised electrochemical chamber and incubated with gentle agitation for 12 min (MCF-7 cells) and 25 min (whole human blood). The surface was gently washed with growth medium (4 mL) three times before being stained with FITC-conjugated anti-EpCAM (5 µg mL$^{-1}$) and Hoechst 33342 (33 µM) by incubating the cells for 20 min. The surface was then gently washed with growth medium 3 times to remove unbound FITC-conjugated anti-EpCAM. Growth medium (6 mL) was placed in the electrochemical chamber before it was placed on the microscope stage. A sterilised Ag|AgCl (3 M KCl) reference electrode and sterilised platinum mesh counter electrode was placed into the growth medium. Phase contrast and fluorescence images were then captured at ×10 and ×40 magnification. The white light and UV lamp was turned off and the white laser was turned on and moved to illuminate a predetermined single captured MCF-7 cell. −1.2 V (vs Ag|AgCl (3 M KCl)) for 240 s was applied to the surface to electrochemically cleave the monolayer below the illuminated cell. The released cell was then recovered using a micromanipulator (Eppendorf InjectMan® 4 fitted with an Eppendorf CellTram® vario and Eppendorf Polar Body Biopsy Tip MML) for downstream analysis of the released cell.

**Synthesis and amplification of cDNA**. Single cells that were electrochemically released and recovered with the micromanipulator were placed in direct lysis buffer (2 µL) in a PCR tube immediately after being released. The reverse-transcription, SPIA and labelling of the amplified cDNA has been described in the Ovation® One-Direct System user guide[37]. Gene expression was measured using an Illumina® Human HT-12 v4 Expression BeadChip with data being analysed using the Partek® Genomic Suite software. For MCF-7 cells, six single released cells and six single untreated cells were collected and hierarchical clusters were graphed for the genes of interest related to epithelial cancer cells[8]. For lung tumour cells, six tumour cells with a high uptake of doxorubicin hydrochloride and six tumour cells with a low uptake of doxorubicin hydrochloride were collected and hierarchical clusters were graphed from gene lists generated by pair-wise comparisons between the cells of varying doxorubicin hydrochloride-uptake with a fold change > 2 or < -2 and an unadjusted $p$-value ≥ 0.05, one-way ANOVA. The expression levels of the tumour cells for genes that are known to be responsible for doxorubicin hydrochloride-uptake were also investigated.

**Data availability**. All relevant data is available from the authors upon request.

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

## Acknowledgements

We would like to thank Tanya Dwarte for providing mouse blood for initial spiking experiments and Dr Joshua McCarroll for providing the H441 NSCLC cell line for subcutaneous injections into mice. We would also like to thank the Mark Wainwright Analytical Centre and the Ramaciotti Centre for Genomics, both at UNSW, for use of their facilities. This work acknowledges funding from the ARC Centre of Excellence in Convergent Bio-Nano Science and Technology (CE140100036), the ARC Laureate Fellowship (FL150100060) program and a National Health and Medical Research Council program grant (1091261).

## Author contributions

S.G.P. developed the capture & release surface chemistry and performed the single-cell gene expression experiments. Y.Y. developed the technology to localise the release of single cells. S.C. developed the synthesis for the electrochemically switchable molecule and provided insights into the electrochemical aspects within the manuscript. B.G. provided guidance and insight into bioconjugating the antibodies to the silicon surface and presenting the biological data. K.K. performed the subcutaneous injection of H441 NSCLC cells into mice before excising the formed tumours. F.M.M. provided the idea of selecting a single cell to release based on the uptake of doxorubicin hydrochloride. M.K. and K.G helped design the biological experiments and assisted in data interpretation. J.J.G. led the project, was involved in all aspects of the project design and data interpretation. S.G.P., Y.Y. and J.J.G. wrote the manuscript with assistance from all authors.

## Additional information

**Competing interests:** The authors declare no competing interests.

