## [Peer Review File · Nature Communications]

Reviewers' comments:

Reviewer #1 (Remarks to the Author):

The manuscript describes a novel and effective procedure for a selective release of surface-captured cells by means of electrochemically cleavable linker molecules assembled on semiconducting silicon (non-patterned!) surface. The specificity of cell release is achieved by light activation and tuned in conjunction with the semiconducting properties of the silicon surface. The concept is interesting and clever, and proved to be very efficient. The cells do not seem to be affected by the developed procedure, as there is no effect on the genetic expression of individually released cells. The platform was used to demonstrate the capabilities in determining the response of each rare cell towards a model drug, including subsequently looking at gene expression of individually released cells. Whole blood (as a 'real' sample) was used to capture rare tumor cells with this approach.

Comment: It would be interesting to know what is the effect of the linker's length (between the silicon surface and the electrochemically cleavable linker) on the potential needed for the release and on the cell viability.

I recommend publishing as is.

Reviewer #2 (Remarks to the Author):

This paper describes an interesting approach whereby electrochemically cleavable surfaces and light activation concepts are combined to enable selection and release of specific cells from the substrate. The paper is well executed and well written, however, the application is not particularly compelling or high impact. It is somewhat difficult to imagine such a complex system being used for circulating tumor cell detection. The authors are some of the leaders in the field of electrochemically switchable surfaces and know that this field has a history of almost two decades. Novelty of the present study is the ability to use light to release specific cells from an electrode surface. This obviates the need for specific electrodes addressing the cells of interest. This concept is interesting and innovative but the application is not particularly compelling. There are photocleavable linkers that can be used release cells from simple optically transparent culture substrates. The need for silicon surfaces and custom-made electrochemically active linkers is not obvious.

So as an academic exercise this paper is interesting and publishable in the chemistry journal where design of the surface may be highlighted. Broad impact of this paper to the biomedical and cancer research community may be somewhat less obvious and my suggestion would be to be publish this paper in a specialized journal.

Reviewer #3 (Remarks to the Author):

The developed platform have shown unique system for the release of specific cell type under the microscopic. The cell release system is based on light and electrochemical stimulus. The authors have discussed the surface modification technique and other methods in details. The method for the capture of specific cell type using antibody affinity is not unique as it has been reported by various groups earlier and wide literature is present. I do not see much novelty in capturing specific cell types from the heterogeneous mixture. However, the release system is unique and novel and may have large scale application when applying on modified surfaces.

The present system is interesting and unique but such methods are laborious & tedious when picking single cell from the modified surface. However, the developed platform can be coupled with

microfluidic platform for better control on single cell recovery after release.

Author should include a recent work by Deng et al; "An Integrated Microfluidic Chip System for Single-Cell Secretion Profiling of Rare Circulating Tumor Cells. Scientific Reports volume 4, Article number: 7499 (2014)". Deng et al have also taken a similar system to validate the capture and release system on integrated microfluidic chip.

The system may fail to release the captured single cell which are present in close proximities because the ligand presentation on surface is not controlled. The developed system may show better result if Antibody(ligand) coupled on micropatterned surface having functional areas separated with significant distances.

The developed platform have potential but only lack the evidences for efficient release and recovery of single cells in quick time for large scale screening and analysis of single cell.

Statistical methods are not considered in all the presented graphs. Errors bars are not direct indications of statistical significance, an appropriate test should be included and p-value should be mentioned in each result. A detailed should be included in method section.

Overall, manuscript is well organized but there are some grammatical errors and long sentences which makes it difficult to understand.

Responses to reviewers' comments

Reviewer #1 (Remarks to the Author):

The manuscript describes a novel and effective procedure for a selective release of surface-captured cells by means of electrochemically cleavable linker molecules assembled on semiconducting silicon (non-patterned!) surface. The specificity of cell release is achieved by light activation and tuned in conjunction with the semiconducting properties of the silicon surface. The concept is interesting and clever, and proved to be very efficient. The cells do not seem to be affected by the developed procedure, as there is no effect on the genetic expression of individually released cells. The platform was used to demonstrate the capabilities in determining the response of each rare cell towards a model drug, including subsequently looking at gene expression of individually released cells. Whole blood (as a 'real' sample) was used to capture rare tumor cells with this approach.

Comment: It would be interesting to know what is the effect of the linker's length (between the silicon surface and the electrochemically cleavable linker) on the potential needed for the release and on the cell viability.

Author reply: We thank the reviewer for their kind words. The redox potential is a thermodynamic quantity and not a kinetic quantity and hence the linker length does not affect the redox potential. This has been demonstrated by Gooding and co-workers on silicon surfaces as used here¹. The following text has been added to the manuscript (line 195)': "The linker length between cleavable layer and silicon surface will not affect the applied potential for cell release as has been demonstrated by Gooding and co-workers²⁵."

Reviewer #2 (Remarks to the Author):

This paper describes an interesting approach whereby electrochemically cleavable surfaces and light activation concepts are combined to enable selection and release of specific cells from the substrate. The paper is well executed and well written, however, the application is not particularly compelling or high impact. It is somewhat difficult to imagine such a complex system being used for circulating tumor cell detection. The authors are some of the leaders in the field of electrochemically switchable surfaces and know that this field has a history of almost two decades. Novelty of the present study is the ability to use light to release specific cells from an electrode surface. This obviates the need for specific electrodes addressing the cells of interest. This concept is interesting and innovative but the application is not particularly compelling. There are photocleavable linkers that can be used release cells from simple optically transparent culture substrates. The need for silicon surfaces and custom-made electrochemically active linkers is not obvious. So as an academic exercise this paper is interesting and publishable in the chemistry journal where design of the surface may be highlighted. Broad impact of this paper to the biomedical and cancer research community may be somewhat less obvious and my suggestion would be to publish this paper in a specialized journal.

Author reply: We thank the reviewer for their report and have revised the manuscript to better highlight the novelty of the paper. As the reviewer pointed out, the purpose of the technology was not to enumerate circulating tumor cells, as there are a number of well-developed methodologies for that already. Tumor heterogeneity is a major issue in cancer therapy and studies of circulating tumour cells have predominantly focused on collection and lysis of cells for genetic analysis with no consideration for biological evaluation². The purpose of the new technology we present in this paper is to meet an unmet challenge; a tool that allows the isolation and further manipulation of rare cells such that heterogeneity in these rare cells can be studied. This is applicable to understanding cell heterogeneity in CTCs which is important for both treatment strategies and fundamental research, rare foetal cells which is important for understanding variations in health exhibited by women before and after giving birth and rare adult stem cells which will enable a better understanding of the variations observed in the efficiency of tissue repair.

Thus, we are aware of many applications for this technology that our clinical collaborators are seeking. The new technology provides this previously unrealisable capability. Although the referee found the surface complex we contend that, if commercially supplied, it is quite simple as all the end user needs is a turnkey microscopy and a turnkey potentiostat. That is, it is in effect a smart microscope slide that fits within the normal workflow of a cell biology lab. The commercial version of the technology will be simpler to operate than many microfluidic devices. The referee is correct in that the current chemistry is

not widely available. However, as the proof-of-principle has worked so well, we have designed a simpler way of fabricating the surface, where the substrate is transparent, which we are currently patenting and discussing with potential commercial partners. Therefore, we feel the method is not only useful for cell biology but will be simpler to use.

The referee mentioned that photocleavable linkers exist that could allow single cells to be released. We are pleased this point was raised; the novelty in our paper is that both light and electrochemical potential are required to release specific cells from an unpatterned surface. This is very important because if light alone was required to release the cells, then visualisation of the surface would cause the cells to automatically be released, and hence the ability to release single cells of interest when desired from a surface of many cells would be lost. The exception would be if the photocleavable linker required UV light but that will then expose the cells to potential genetic damage from light of that energy. We have attempted to reiterate these points more strongly by making the following adjustments to the manuscript:

Add "For example, if these single-cell isolation techniques were used to further understand the functional effects of the rare adult stem cells, rare foetal and maternal cells or rare CTCs within a complex sample, the unsynchronised nature of the much more abundant contaminating cells would hide any functionally relevant information obtained from the rare cells within the sample." on line 58

Change "To then explore the heterogeneity of the cells requires individual cells to be addressed" to "To then explore the heterogeneity of these rare cells requires them to be addressed individually" on line 68.

Add "If these surfaces were used with the rare cells, then the further exploration could only be possible on an ensemble number of rare cells. Performing the further analysis on the stem cells, for example, would highlight the potential reasons for the observed variation in tissue repair but it would not reveal whether these differences are as a result of an equal contribution of all cells within the population or are dominated by a select few cells within the population. For this reason, it would be advantageous to be able to release only one cell. To control the release of a single cell however, is a considerable challenge that requires two stimuli so that the cell can first be observed and then released when desired. Herein we use a combination of light and electrochemical potential. Using light alone to release a single cell would mean visualising the cell would also release it. Using potential alone would not confine the release to a single cell. As such we visualise the surface and then reduce the beam size of the light to just one cell where upon potential is applied and a single cell releases (Fig. 1a)." on line 73.

Delete "The challenge is how to first observe the surface and then confine the release stimulus to a single cell of one's choosing. Light alone could not be used as observing the surface with a microscope would release all cells or, if UV and shorter wavelengths of light were used, there is a risk of damaging the cells." After "Using potential alone would not confine the release to a single cell. As such we visualise the surface and then reduce the beam size of the light to just one cell where upon potential is applied and a cell releases (Figure 1a)" on line 85.

Change "The purpose of this paper is to present a novel 'capture & release surface'" to "The purpose of this paper is to present the development of this novel 'capture & release surface'" on line 85.

Add "1)" before 'capture rare cells on the surface using antibodies for cell-surface markers' on line 86.

Add "2)" before 'allow the surface to be imaged with a microscope' on line 87.

Change "then any single cell on that surface can be released" to "3) then release any single cell on that surface when desired" on line 87.

Delete "To achieve the ability to image the captured cells and then release a single cell necessitated the development of a release surface that would only release the cell of interest upon application of two stimuli, light and electrochemical potential." After "3) then release any single cell on that surface when desired" on line 88.

Add "This means that preliminary information about the heterogeneity amongst rare cells, such as drug response, can be obtained before selecting the single cell of interest to be released and further explored." after "In this way, a completely unpatterned surface can be prepared for the capture of the rare cells but individual cells can be released, when, and only when, two stimuli are applied to initiate the release; light is confined to a given location and application of the appropriate potential (Fig. 1A)" on line 96.

To summarise our response to this point, we would like to highlight that that this was a proof-of-concept study with the most well-defined surface we could make to show that light activated

electrochemistry can be used to interrogate rare cells before isolating individual cells for further analysis. We believe that the ability to observe heterogeneity amongst rare cells, particularly with regards to the response of individual cells towards a treatment, prior to selecting an individual cell to isolate to further understand origins to this variation in response, is novel and a useful tool for cell biology. Since the approach has worked so well, we have designed a simpler way of fabricating the surface which we are currently patenting and discussing with potential commercial partners.

Reviewer #3 (Remarks to the Author):

The developed platform have shown unique system for the release of specific cell type under the microscopic. The cell release system is based on light and electrochemical stimulus. The authors have discussed the surface modification technique and other methods in details. The method for the capture of specific cell type using antibody affinity is not unique as it has been reported by various groups earlier and wide literature is present. I do not see much novelty in capturing specific cell types from the heterogeneous mixture. However, the release system is unique and novel and may have large scale application when applying on modified surfaces.

Author reply: We agree with the reviewer that while the capturing of a specific cell type is not novel, the ability of the device to first capture the rare cells, then interrogate the rare cells to find and realise unique single cells amongst this is the major strength of our paper. We have revised the manuscript to emphasise this novelty in response to Reviewer 2.

The present system is interesting and unique but such methods are laborious & tedious when picking single cell from the modified surface. However, the developed platform can be coupled with microfluidic platform for better control on single cell recovery after release.

Author reply: We would like to highlight that that this was a proof of concept study with the most well-defined surface we could make to show that light activated electrochemistry can be used to interrogate rare cells before isolating individual cells for further analysis. Since it has worked so well, we have designed a simpler way of fabricating the surface over which we are currently patenting and in discussion with potential commercial partners. Microfluidics could be adopted to further simplify this process.

“Microfluidics could also be coupled with this device to improve the speed to isolate a single cell” has been added after ‘Hence the behaviour of each cell on the surface can be determined prior to selecting a unique cell to release, which could then be further characterised or cloned to better understand its relative functional contribution to the progression of the disease’ on line 376.

Author should include a recent work by Deng et al; "An Integrated Microfluidic Chip System for Single-Cell Secretion Profiling of Rare Circulating Tumor Cells. Scientific Reports volume 4, Article number: 7499 (2014)". Deng et al have also taken a similar system to validate the capture and release system on integrated microfluidic chip.

Author reply: We would like to thank the reviewer for this recommendation and have rephrased the text and included the references on lines 54 and 73. It now reads: “Examples include flow cytometry, micromanipulation or encapsulating single cells within a microwell, water droplet or a dielectrophoretic cage^{2,3,9,10}” (see line 54).

The system may fail to release the captured single cell which are present in close proximities because the ligand presentation on surface is not controlled. The developed system may show better result if Antibody(ligand) coupled on micropatterned surface having functional areas separated with significant distances.

Author reply: We have included a video highlighting the capability of the device to release a single cell, even when a neighbouring cell is only 30 μm way: “Video S3 shows that a spatial resolution of 30 μm can be obtained with this device.” (line 269).

The developed platform have potential but only lack the evidences for efficient release and recovery of single cells in quick time for large scale screening and analysis of single cell.

Author reply: We would like to thank the reviewer for this comment and recognize that we have not been clear enough within the manuscript. We believe that the technology is not designed for large scale screening as good technologies, such as FACS, already exist for this purpose. However, with these technologies, discerning rare, but important events from noise is challenging. The novelty of our device is

the ability to examine heterogeneity amongst rare cells. We have attempted to reiterate this point (see response to Reviewer 2). Furthermore, we want to emphasise that this is a proof-of-concept study. The speed of release can be improved by swapping from the use of a light source to activate the silicon surface that is controlled with a stepping motor to a light that is controlled with a crystal projector. The speed of recovery can be improved through the use of microfluidics.

Statistical methods are not considered in all the presented graphs. Errors bars are not direct indications of statistical significance, an appropriate test should be included and p-value should be mentioned in each result. A detailed should be included in method section.

Author reply: We have now included a statistical analysis:

“Error bars indicate 95% confidence interval, n = 6” (line 147)

“Error bars indicate 95% confidence interval, n = 6” (line 211)

“Error bars indicate 95% confidence interval, n = 6” (line 214)

“(when compared to the IC₅₀ values before capture & release for different drugs)” (line 237).

“Error bars indicate 95% confidence interval, n = 6” (line 298)

“Statistical analysis

Uncertainties depict the 95% confidence interval and were calculated using $\mu_{95\%} = s \cdot t_{0.025} / \sqrt{n}$ where the degrees of freedom = n - 1. Comparison of two treatments, such as cultured vs captured & released was quantitated by setting up a null hypothesis, H₀, that the treatments were the same ($x_1 - x_2 = 0$) and an alternate hypothesis H_a, that the treatments were not the same ($x_1 - x_2 \neq 0$) where x_1 = the larger average - $\mu_{95\%}$ and x_2 = the smaller average + $\mu_{95\%}$.” after line 454.

Overall, manuscript is well organized but there are some grammatical errors and long sentences which makes it difficult to understand.

Author reply: We thank the referee for the basic but significant point, corresponding changes (such as lines 22-30, lines 42-46, lines 92-98, lines 192-195, and lines 390-393) have been made in the manuscript and supplementary information.

Reference

1. Darwish N, Paddon-Row MN, Gooding JJ. Surface-Bound Norbornylogous Bridges as Molecular Rulers for Investigating Interfacial Electrochemistry and as Single Molecule Switches. *Acc Chem Res* **47**, 385-395 (2014).
2. Hunter KW, Amin R, Deasy S, Ha N-H, Wakefield L. Genetic insights into the morass of metastatic heterogeneity. *Nat Rev Cancer* **18**, 211-223 (2018).

REVIEWERS' COMMENTS:

Reviewer #1 (Remarks to the Author):

The authors addressed well the reviewers' comments, which improved the manuscript. However, a similar concept of cells release from a semiconducting surface using an electrochemically cleavable linker has been published recently (and is not cited): "Dynamic Poly(3,4ethylenedioxythiophene)s Integrate Low Impedance with Redox-Switchable Biofunction" by Hsing An Lin, Bo Zhu Yu-Wei Wu, Jun Sekine, Aiko Nakao, Shyh-Chyang Luo, Yoshiro Yamashita, Hsiao-Hua Yu, Adv. Funct. Mater. doi.org/10.1002/adfm.201703890

The authors now need to cite and carefully discuss the above-mentioned paper, and outline the differences and novelty of their work in regard to the published work by Lin et al.

Reviewer #3 (Remarks to the Author):

Authors have answered all the comments and incorporated suggested changes. After reading their revised manuscript i can realize that the work is only a proof of concept for separating and analyzing cells based on photoelectrochemical platform with a very specific application. However, I may recommend this manuscript for publication due to novel cell release system which is interesting.

Responses to referees' comments

Reviewer #1 (Remarks to the Author):

The authors addressed well the reviewers' comments, which improved the manuscript. However, a similar concept of cells release from a semiconducting surface using an electrochemically cleavable linker has been published recently (and is not cited):

"Dynamic Poly(3,4ethylenedioxythiophene)s Integrate Low Impedance with Redox-Switchable Biofunction" by Hsing An Lin, Bo Zhu Yu-Wei Wu, Jun Sekine, Aiko Nakao, Shyh-Chyang Luo, Yoshiro Yamashita, Hsiao-Hua Yu, Adv. Funct. Mater. doi.org/10.1002/adfm.201703890

The authors now need to cite and carefully discuss the above-mentioned paper, and outline the differences and novelty of their work in regard to the published work by Lin et al.

Author reply: The authors appreciate the suggestion and have added this reference on line 75. The technology within this cited paper does describe a redox-responsive switchable surface but is only capable of releasing an ensemble number of cells, due to the ITO not being able to be light-activated. Introduction to this reference helps to re-iterate the novelty of the current work within this paper, and better clarify the ability of the system to observe a specific rare cell before selecting to recover it from other rare cells.